# Optimizing Maritime Search and Rescue Planning via Genetic Algorithms: Incorporating Civilian Vessel Collaboration

**DOI:** 10.3390/biomimetics10090588

**Published:** 2025-09-03

**Authors:** Seung-Yeol Hong, Yong-Hyuk Kim

**Affiliations:** Department of Computer Science, Kwangwoon University, 20 Kwangwoon-ro, Nowon-gu, Seoul 01897, Republic of Korea; hong7117y@gmail.com

**Keywords:** search and rescue planning, genetic algorithm, greedy algorithm, civilian cooperation

## Abstract

This study proposes a biomimetic optimization approach for maritime Search and Rescue (SAR) planning using a Genetic Algorithm (GA). The goal is to maximize the number of detected drifting targets by optimally deploying both official and civilian Search and Rescue Units (SRUs). The proposed method incorporates a POD-adjusted fitness function with collision-avoidance constraints and is enhanced by a greedy initialization strategy. To validate its effectiveness, we compare the GA against a baseline method (EAGD) that combines a (1 + 1)-Evolutionary Algorithm with greedy deployment, across 24 experiments involving 2 realistic maritime scenarios and 12 coverage conditions. Results show that GA consistently achieves higher average fitness and stability, particularly under stress-test settings involving only civilian vessels. The findings underscore the potential of biomimetic algorithms for real-time, flexible, and scalable SAR planning, while highlighting the value of civilian participation in emergency maritime operations.

## 1. Introduction

### 1.1. Motivation

Maritime Search and Rescue (SAR) planning establishes a plan to detect and rescue drifting human lives, damaged vessels, and material assets caused by maritime disasters and accidents. Given that its primary objective is to save human lives, SAR planning is critically important and must be conducted quickly and accurately to increase the survival rate of rescue targets [1].

SAR planning includes selecting Search and Rescue Units (SRUs), deploying search areas to the selected SRUs, and generating corresponding search paths. The process of selecting SRUs constitutes a combinatorial optimization problem, while deploying search areas to the space constitutes a spatial optimization problem. Both are categorized as NP-Hard problems, making it difficult for humans to manually establish efficient plans. In particular, with the advancement of technology, the increasing variety and number of SRUs and the growing scale and complexity of the search area further highlight the limitations of manual planning.

Accordingly, various systems have been developed to automate or assist in generating SAR plans. For example, the United States Coast Guard has operated the Search and Rescue Optimal Planning System (SAROPS) since 2007, which integrates Monte Carlo-based particle filters with high resolution environmental data to automate drift prediction and SRU resource allocation [2]. Breivik and Allen proposed a drift prediction technique using a Bayesian-based particle filter for the Norwegian maritime search system [3].

Recently, studies applying genetic algorithms (GAs) and swarm-based metaheuristics to SAR resource allocation and path optimization have been actively conducted. Zhang et al. proposed the GSAA method, which combines GA and simulated annealing, and reported higher rescue success rates compared to conventional methods in the Bohai Sea, China [4]. Dong et al., based on the South China Sea case, adopted a bi-objective optimization model using NSGA-II to balance resource allocation and efficiency [5]. Yoo and Kim presented an optimal deployment method for rescue vessels in archipelagic regions by combining dynamic programming and particle swarm optimization (PSO), also reporting execution efficiency [6]. Other approaches include a two-stage integer programming-based SAR resource allocation integrating GA and PSO [7], a time-space weighted PSO for helicopter SAR decision making [8], GA-RL-based dynamic response scheduling that fuses reinforcement learning and GA [9], a DE-based multi criteria decision making model [10], and an APSO-GA approach based on clustering accident-prone areas [11].

However, many existing studies in maritime SAR optimization primarily focus on official SRUs owned by professional maritime search organizations (e.g., Korea Coast Guard, United States Coast Guard, and Canadian Joint Rescue Coordination Centre), with relatively limited consideration of civilian SRU participation potential. In recent maritime accidents, the voluntary participation of civilian vessels has emerged as a key factor in increasing SAR success rates. Stoyanov et al. emphasized the role of civilian vessels, stating, “The global search and rescue (SAR) system heavily relies on merchant cargo vessels for operations of assistance and rescue at sea” [12].

Moreover, recent studies highlight the increasingly significant role of civilian volunteers and non-governmental organizations (NGOs) in filling critical gaps in maritime search and rescue (SAR) operations, particularly in high-volume crisis zones such as the Mediterranean Sea. In scenarios where official assets are delayed, insufficient, or politically constrained, these civilian actors often undertake frontline rescue missions using privately owned vessels. For instance, NGOs such as and the Alan Kurdi/RESQ PEOPLE fleet have conducted numerous operations rescuing migrants in distress, frequently under urgent and resource-limited conditions. These efforts emphasize that the active involvement of civilian vessels is a critical component of modern SAR operations, directly enhancing success rates and shaping the way future optimization approaches should integrate both official and civilian SRUs [13].

Therefore, this study proposes an optimization approach for SAR planning that determines the search areas of civilian SRUs effectively using a GA and optimizes SAR plans based on the integration of official and civilian SRUs. This approach is designed to provide a more flexible and scalable search strategy for SAR planning, while also exploring the potential of a biomimetic optimization approach that adapts to complex environments and efficiently allocates resources.

### 1.2. Genetic Algorithms

The SAR planning problem resembles environmental adaptation in natural ecosystems, as both involve allocating limited resources under dynamic and uncertain conditions. For this reason, this study employs the Genetic Algorithm (GA), a representative biomimetic metaheuristic that mimics the evolutionary process in nature [14,15].

GA simulates the evolutionary process in which individuals with higher adaptability to the environment leave more offspring and evolve over generations, based on the principle of survival of the fittest. The core operators of GA, selection, crossover, mutation, and replacement correspond to natural survival competition, gene exchange, genetic mutation, and generational replacement, respectively. The selection operator increases the probability that individuals with high fitness are chosen as parents, preserving superior genes. The crossover operator combines the genes of parents to generate new solutions. The mutation operator maintains solution diversity to prevent convergence to a local optimum, while the replacement operator forms new generations to continue the search process.

Due to this evolutionary search structure, GA can efficiently explore wide and complex search spaces, which aligns structurally with the SAR problem, where optimal rescue paths must be progressively searched for under various constraints and dynamic factors of maritime areas and SRU resources. Therefore, GA is evaluated as a suitable biomimetic optimization approach for automating and optimizing SAR planning.

## 2. Materials and Methods

### 2.1. Input Data

This section describes the problem setting and the input data. Although not addressed in our study, SAR planning requires the prior step of determining the location of the search target. To identify the location, the search target is represented as drift particles, which are placed at the accident site, and their drift trajectories are predicted. Recent studies employ various artificial intelligence models for predicting drift trajectories [16,17,18]. Nevertheless, due to the inherent complexity of marine environments, accurately predicting the drift trajectories of drift particles is very challenging, even with artificial intelligence. Therefore, modern SAR planning places multiple drift particles (approximately 1000–4000) on the initial drift locations near the accident site and predicts the drift trajectories of each drift particle. Even when drift particles are placed at the same location, their drift trajectories differ due to the seed value of the artificial intelligence model or slight differences in initial positions Figure 1). Among these widely dispersed drift particles, it is assumed that the actual search target exists at one of their positions. Hence, the objective of the search planning phase is to maximize the probability of success by including as many drift particles as possible.

Meanwhile, SRUs typically conduct searches using parallel sweep patterns [19]. Thus, the SRU search area is generally rectangular; in our study, all SRUs are assumed to cover square search areas. The size of the search area is constrained by the SRU’s capability and available search time. The factors determining the search area size are the SRU’s search speed (*V*), detection range (*W*), and search time (*T*). Therefore, the area of an SRU’s search coverage is expressed as A=V×T×W, and since the search area is set as a square, the SRU’s search coverage becomes a square with side length $\sqrt{A}$.

Accordingly, we use two inputs: (1) a map indicating the drift particle (search targets) positions (Particle data), and (2) information on SRU search capabilities and search times (SRU data). This forms the basic experimental environment. An example of the input data is shown in Figure 2. Figure 2a represents the drift particle position map, where the latitude and longitude of drift particles are provided. These positions are projected onto a two-dimensional plane using the Mercator projection [20]. Figure 2b shows SRU information and search times. The required SRU information includes the name, search speed, detection range, search altitude, and search time. The reason for including search altitude is explained in Section 2.2.

Although each scenario is anchored to a real accident case, the drift trajectories used in our experiments are simulated via a machine learning drift model conditioned on environmental/forecast fields; thus, the dataset is simulation-based but incident-realistic. The pipeline is data-agnostic and can be applied without modification to observational as well as physics-based simulated datasets.

### 2.2. Genetic Algorithm Design for SRU Deployment

This section explains how we apply a genetic algorithm to our problem. First, it is necessary to design a chromosome suitable for the problem. We deploy search areas (modeled as rectangles) on a two-dimensional plane representing the sea. Since we predefine the search areas as square shapes, the only information required to represent each search area is its position. Thus, our chromosome consists of the *x* and *y* coordinates of the square’s center point. When deploying *k* SRUs, the search areas also total *k*, so each chromosome is represented as a one-dimensional array containing *k* center coordinates of squares. Under this chromosome design, deploying a single search area poses a problem in two-dimensional space. When deploying *k* search areas, the decision space has dimension 2k. For example, when 35 SRUs are deployed, the problem dimension is 70, while for 66 SRUs, it becomes 132. Across our experiments, the dimensionality ranged from approximately 70 to over 130, depending on the scenario and coverage level.

As described in Section 2.1, the objective is to deploy search areas to cover as many drift particles as possible, and the fitness function must reflect this objective. The fitness function F(S) is defined as Equation (Equation 1). The goal is to maximize the fitness function F(S).(1)$$F\left(S\right)=\sum_{k=1}^{\left|P\right|}I\left(p_{k}\in S\right)$$
where *S* denotes the set of search areas represented by the chromosome, |P| denotes the number of drift particles, and $p_{k}$ represents the *k*-th drift particle.

Additional constraints are introduced to ensure applicability in real-world scenarios. In operational settings, overlapping search areas may lead to collision risks among SRUs, leading to secondary incidents. The risk of such collisions depends on the types of SRUs operating in the same region. For example, helicopters and vessels can conduct coordinated searches over the same area when properly managed, whereas multiple vessels operating in overlapping regions pose higher collision risks. According to the IAMSAR Manual, altitude separation between aircraft is generally set to approximately 150–300 m (500–1000 ft) to reduce the likelihood of collision [1]. Based on this guideline, we incorporate altitude separation as a safety constraint in the fitness function.

Additionally, we impose a slight penalty when search areas operating at different altitudes overlap. Without such a penalty, search areas would be concentrated only in regions with dense drift particle distributions, hindering broader coverage. This violates our objective of covering as many drift particles as possible. The penalty is designed based on the Probability of Detection (POD) formula proposed by Xiong [21]. The POD refers to the probability that a drift particle within a search area is detected by an SRU. When counting drift particles within a search area, we calculate the POD. If a drift particle is counted multiple times due to overlapping search areas, its contribution is adjusted based on the *p*-*fail* value [22]. The *p*-*fail* value is calculated by multiplying the probability of a previous SRU failing to detect the drift particle by the current SRU’s POD value and adding it to the existing POD. Equation (Equation 2) shows how to compute the POD $\lambda_{k}$ of the *k*-th drift particle by applying the *p*-*fail*.(2)$$\lambda_{k}=\sum_{i=1}^{\left|R\right|}\lambda \left(r_{i}\right)\prod_{j=0}^{i-1}\left(1-\lambda (r_{j})\right)$$
where *R* denotes the set of search areas that contain the *k*-th drift particle, $r_{i}$ represents the *i*-th search area in *R*, and λ(r) denotes the POD of search area *r*, defined as $\lambda \left(r\right)=1-e^{-C}$. The coverage factor *C* is computed as the ratio of the detection range (*W*) to the track spacing. By default, *C* is set to 1, but it can be adjusted depending on the experimental setting.

The track spacing refers to the distance between adjacent legs in the parallel sweep pattern used by an SRU during search operations. Intentionally reducing the track spacing decreases the search area *A* of the SRU, but enables denser coverage of a smaller region, thereby increasing the POD P(r).

Finally, Equation (Equation 3) defines the final form of the fitness function, which incorporates both the collision-avoidance constraint and the *p*-*fail*-based detection adjustment.(3)$$\begin{array}{cc} F(S) & =\sum_{k=1}^{\left|P\right|}D\left(S,\;p_{k}\right)\\ \mathrm{subject}\mathrm{to}\\ \;D(S,\;p_{k}) & =\left\{\begin{array}{cc} P_{k}, & \mathrm{if}\;|a\left(r_{i}\right)-a\left(r_{j}\right)|\geq \alpha ,\;\mathrm{for}\;\mathrm{all}\;r_{i},\;r_{j}\in R\\ 0, & \mathrm{otherwise} \end{array}\right. \end{array}$$
where $a(r_{i})$ denote the search altitude of the *i*-th search area, and α denotes the minimum altitude separation between SRUs. In this study, we set α=150 m; this parameter may be adjusted according to SRU capabilities and operational/environmental conditions (e.g., weather).

The genetic algorithm operators used in our research are as follows. For the selection operator, we use Roulette-wheel selection [23,24]. Roulette-wheel is designed to increase the probability of selecting chromosomes with high fitness while still allowing those with low fitness to be selected probabilistically. By enabling the selection of chromosomes with low fitness, the solution search space can be expanded, allowing chromosomes with low overall fitness but possessing beneficial genetic characteristics to be preserved.

For the crossover operation, Uniform crossover is employed [25]. Uniform crossover selects genes from parent chromosomes according to a probability *p* and passes them to offspring. However, if each gene is independently inherited, the problem space becomes too large, and offspring chromosomes may fail to inherit well-positioned search areas. For example, if the first search area of parent 1 is located at (50, 40) (favorable position) and parent 2 at (30, 90) (unfavorable position), a naïve uniform crossover might generate (50, 90), losing the beneficial configuration. To preserve the spatial characteristics of search areas, we probabilistically select the entire center coordinates of search areas from one parent as a unit rather than mixing *x* and *y* coordinates separately. In our study, we set p=0.5.

The mutation operation slightly moves search areas to explore new chromosomes. The movement distance varies according to *x* and *y* coordinates, defined as half the length of the entire map’s *x*-dimension ($x_{half}$) and *y*-dimension ($y_{half}$), respectively. Instead of selecting movement distances uniformly, we apply a normal distribution with mean 0 and standard deviation $x_{half}$ for *x* and mean 0 and standard deviation $y_{half}$ for *y*. This strategy prevents drastic displacements of well-positioned search areas while still permitting occasional larger moves to escape local optima.

The repair operation corrects infeasible chromosome configurations produced by crossover or mutation. Infeasible solutions occur when search areas move outside the entire map. In such cases, the search area is shifted to the nearest map boundary to restore validity.

For the replacement operation, we adopt an elitism strategy, where offspring chromosomes replace only parent chromosomes with lower fitness, thus preserving the best-performing individuals in the population [24,26].

To substantiate the performance of our genetic algorithm, we conduct experiments across diverse parameter combinations. Based on preliminary testing, we selected the following configuration: 2000 generations, population size of 700, 70 offspring per generation, and 30% mutation rate. The computational complexity of fitness evaluation scales as O(k·|P|) per generation, where *k* denotes the number of SRUs and |P| represents the number of drift particles, ensuring tractability even for large-scale SAR instances.

Figure 3 shows the convergence behavior for the best-performing configuration using the stress-test dataset (Case 1, Scenario 3, coverage ratio 1.0; details in Section 3.1). Each point represents the average fitness of the population in that generation. The convergence curve indicates that the algorithm stabilizes at approximately generation 1000. Based on this analysis, we adopt the following parameters for all subsequent experiments: 1000 generations, population size 700, 70 offspring per generation, and 30% mutation rate.

### 2.3. Greedy Algorithm

To enhance convergence, we designed a greedy algorithm. Unlike previous studies, our problem involves deploying a significantly large number of search areas. This complexity can hinder the effective convergence of genetic algorithms. To address this, we aim to generate high-quality initial chromosomes that guide the genetic algorithm toward better convergence. For this purpose, we adopt a greedy algorithm that is both fast and capable of producing promising initial chromosomes.

The greedy algorithm we designed operates as illustrated in Figure 4. First, the largest search area is randomly deployed on the map. Then, the remaining search areas are deployed one by one in descending order of size, adjacent to previously deployed search areas. Each new search area is deployed to align one of its edges with an edge of the previous search area. For each edge, left, right, top, and bottom, there are two possible alignment options. For example, if the new search area is deployed along the left or right edge of the previous one, it can be aligned either by matching their bottom edges or their top edges. Similarly, if it is deployed along the top or bottom edge, it can be aligned by matching their left or right edges. This results in a total of 4 edges × 2 alignments = 8 possible deployment configurations. Among these, we evaluate the fitness of each option and select the one with the highest fitness. The process continues until all search areas are deployed.

### 2.4. Comparison Methods

To validate our genetic algorithm, we compare it with a baseline method that we also designed, called EAGD, which combines a (1 + 1)-Evolutionary Algorithm ((1 + 1)-EA) with a greedy algorithm. The (1 + 1)-EA is a simple yet powerful baseline method widely analyzed in evolutionary computation literature [27]. It maintains a single chromosome and iteratively applies mutation—if the resulting offspring has improved fitness, it replaces the current chromosome; otherwise, it is discarded. This process enables gradual improvement while avoiding premature convergence.

EAGD is inspired by field strategies in which the entire search region is divided into *N* equally sized subregions and SRUs are deployed separately within each. This concept is consistent with operational guidelines described in Compatibility of Land SAR Procedures with Search Theory [28]. By partitioning the search area into smaller subregions, EAGD effectively reduces the complexity of the original problem and enables more tractable optimization.

The algorithm proceeds as follows:
Step 1:Divide the entire search area into *N* equally sized sub-regions.Step 2:For each subregion, generate a candidate set of search areas.Step 3:For each subregion:
(a)Use a (1 + 1)-EA to deploy the largest search area, focusing on maximizing fitness.(b)Deploy the remaining search areas using a greedy algorithm explained in Section 2.2.

This approach is directly inspired by practical SAR operations, where the entire search area is typically decomposed into multiple subregions for systematic coverage. By integrating the metaheuristic (1 + 1)-EA with a greedy algorithm, EAGD effectively balances global exploration and efficient local optimization. As it reflects methods already adopted in real-world practice, EAGD serves as a highly appropriate and convincing baseline while still providing a strong and effective optimization framework.

### 2.5. Local Optimization Algorithm

Although the genetic algorithm (GA) generally produces high-quality solutions, it often struggles with fine-grained adjustments, such as eliminating minor overlaps between search areas operating at the same altitude. To address this limitation, we apply a local optimization procedure as a post-processing step to refine the GA output. Although other metaheuristics can be used to fine-tune the GA’s output, combining the GA with additional metaheuristics incurs excessive runtime. In particular, the GA already yields strong solutions, so extensive fine-grained tuning is unnecessary. Therefore, we design a lightweight local algorithm that runs briefly (in 1 s) and performs only minimal adjustments. Specifically, resolving minor overlaps among search areas at the same altitude.

This local optimization is triggered when two search areas with the same altitude overlap. Let us denote the overlapping pair as search areas A and B. We consider five candidate operations to resolve the conflict:(i)Move search area A in either the *x* or *y* direction, whichever requires less distance, until it no longer overlaps with search area B.(ii)Move search area B using the same method.(iii)Move both search areas A and B equally in opposite directions to resolve the overlap.(iv)Reduce the track spacing of search area A, thereby shrinking its coverage area but increasing its POD value.(v)Reduce the track spacing of search area B.

Each of the five candidate solutions is evaluated using the same fitness function described in Section 2.2, which accounts for POD contributions and penalties for overlapping coverage. The operation that yields the highest fitness is selected and applied. This local optimization step is repeated until no further improvement is observed or all overlaps are resolved.

By incorporating this refinement process, we are able to effectively remove minor overlaps while preserving or improving the overall coverage quality.

## 3. Results

### 3.1. Experimental Data

We conduct experiments using three particle datasets, each representing a distinct maritime search scenario. In order of scenarios, the total search area increases. Scenario 1 simulates a night-time search operation, during which the detection range of SRUs is reduced due to limited visibility. As a result, although Scenario 1 covers a smaller geographical area than Scenario 2, it requires more SRUs to achieve the same coverage level. Scenario 3 is a dataset generated from a real incident. Scenarios 1 and 2 also generate drift particles from the same incident, but the searches occur at hypothetical times. In contrast, Scenario 3 determines the search time by accounting for the incident detection time and matches the actual period during which the Korea Coast Guard conducts the search. Therefore, Scenario 3 represents the most realistic simulation dataset.

In addition, we define 18 SRU datasets, categorized into 6 groups according to their total coverage rate-ranging from 50% to 100% in 10% increments. As the required coverage increases, the number of SRUs increases accordingly. For example, in Scenario 1:Under Case 1 (civilian-only deployment), 35 SRUs are required to achieve 50% coverage and 66 SRUs for 100% coverage.Under Case 2 (official-first deployment), only 18 SRUs are needed for 50% coverage and 35 SRUs for 100% coverage.

We evaluate two deployment strategies for assigning SRUs:Case 1—Civilian-only: All search tasks are assigned exclusively to civilian SRUs. This represents a stress-test scenario, intended to evaluate the robustness of our algorithm in the absence of formal SAR units.Case 2—Official-first: Official SRUs are deployed first, and civilian SRUs are used to cover the remaining uncovered areas. This scenario reflects a more realistic operational setting, where coordination between official and civilian resources is essential.

Since the official SRUs cover part of the search area in Case 2, the number of required civilian SRUs is always smaller than in Case 1 under the same coverage condition.

In Case 2, official SRUs are predeployed to the search area before running any optimization algorithm. We utilize our previous work [29] to determine the initial deployment of official SRUs, and the corresponding search areas are fixed throughout the subsequent GA, EAGD, and local optimization. These predeployed search areas are not modified during the optimization process.

However, the fitness function fully incorporates these fixed official search areas. In other words, the fitness evaluation of new solutions (civilian-SRU deployment) accounts for overlaps and interactions with the predeployed official SRUs. This design enables the algorithms to adapt their deployment strategies accordingly.

For example, if an official SRU is searching at an altitude of 0 m, deploying a civilian-SRU search area on top of it results in a significant fitness penalty due to collision risk. Consequently, EAGD and GA tend to avoid deploying search areas of civilian SRUs in overlapping positions. Conversely, if the official SRUs are operating at a higher altitude (e.g., 150 m), some degree of overlap may be acceptable. In such cases, the algorithm may allow minor overlaps to increase the overall fitness, accounting for the adjusted POD and *p*-*fail* values. This behavior is illustrated in Figure 5. In this figure, light red and light blue search areas represent preassigned official SRUs: the light red indicates units operating at 0 m altitude, and the light blue represents units at 150 m altitude. The dashed lines show the six subregions defined by EAGD. Dark red search areas correspond to the search areas assigned to civilian SRUs by EAGD.

In the middle subregion, EAGD tends to avoid overlapping with low-altitude official SRUs (light red), while allowing partial overlap with high-altitude SRUs (light blue) to improve fitness under the *p*-*fail* mechanism. In contrast, in the top subregion, the algorithm completely avoids overlapping with both official SRU types, suggesting that in this specific case, non-overlap yields a better fitness outcome. This example demonstrates how EAGD dynamically adapts its placement strategy in response to altitude-based collision penalties and POD trade-offs.

### 3.2. Experimental Results

Table 1 presents the performance comparison between the proposed GA and EAGD across six coverage ratios under two deployment strategies (Cases 1 and 2) and three scenarios (Scenarios 1, 2, and 3). Performance is evaluated using three metrics: best fitness, average (Ave) fitness, and standard deviation (SD) of fitness values over 30 independent runs, with statistical significance assessed through Welch’s *t*-tests [30].

The results demonstrate that GA consistently achieves superior performance compared to EAGD across all experimental configurations. GA obtains higher best fitness values in every test case, with performance improvements ranging from approximately 1% to 10% depending on the scenario and ratio. In most configurations, EAGD’s best performance falls below GA’s mean performance, indicating the robustness and reliability of the proposed approach. Moreover, Welch’s *t*-tests yielded *p*-values < 0.001 across all experiments, providing strong statistical evidence that the performance differences are highly significant.

On the stress-test dataset (Case 1, Scenario 3, coverage ratio 1.0; deploy 96 search areas), the GA runtime is approximately 20 min. While GA requires higher computational time compared to EAGD, the algorithm remains operationally feasible for SAR planning applications. Given that SRU dispatch preparation typically requires approximately 30 min in practice [1], GA’s computational overhead is acceptable considering its superior solution quality. Additionally, the computational time scales with the number of SRUs, making the algorithm more efficient for smaller, more typical SAR scenarios.

GA demonstrates consistent performance with relatively low standard deviations across most configurations, indicating stable convergence behavior. The algorithm shows particular strength in high coverage conditions and maintains effectiveness across both civilian-only (Case 2) and mixed deployment scenarios (Case 1). These results validate the effectiveness of the proposed greedy initialization strategy and POD-adjusted fitness formulation, confirming GA’s suitability for real-world SAR planning applications.

Figure 6 depicts the mean performance across all experiments as a bar chart. In every dataset, the GA achieves higher fitness than EAGD.

## 4. Discussion

This study demonstrates the potential of biomimetic optimization through the successful application of a Genetic Algorithm (GA) to a highly complex real-world-inspired maritime Search and Rescue (SAR) planning problem. In contrast to previous studies that dealt with relatively small-scale or simplified search configurations, our work tackles the challenge of placing a large number of search areas (modeled as rectangles) under dynamic constraints such as collision risk, altitude-based restrictions, and probabilistic detection.

The structural similarity between the SAR planning task and biological evolution is a key insight driving this study. Just as natural ecosystems must adaptively allocate limited resources across a complex and changing environment, the proposed GA continuously evolves its population to identify increasingly effective SAR deployment plans. This highlights the core strength of biomimetic algorithms: the ability to adapt under uncertainty, interact with constraints, and evolve robust solutions over time.

To further enhance convergence in this high-dimensional search space, we proposed a greedy algorithm that constructs initial solutions by sequentially aligning search areas using an edge-matching strategy. This greedy method not only serves as an effective initialization tool for GA, but also produces standalone high-quality solutions, demonstrating its utility as a lightweight deployment strategy in time-sensitive operations. In particular, scalability was confirmed through a new large-scale case (Case 3) based on a real maritime accident, where deploying 96 SRUs required approximately 20 min, still within the 30 min operational preparation window recommended in SAR practices. This indicates that the greedy initialization strategy remains feasible and effective even in large-scale, stress-test scenarios.

Our experimental results indicate that the proposed GA consistently outperforms baseline methods in terms of coverage and robustness, even under stress-test scenarios involving only civilian SRUs. This is particularly noteworthy given the increasing importance of civilian and volunteer participation in modern SAR operations. While many prior works focused solely on official SRUs, our method explicitly incorporates civilian cooperation as a strategic asset, offering flexible and scalable deployment plans that adapt to available resources. As a baseline, we employed the Efficient Area-Greedy Deployment (EAGD), which reflects a practically used strategy of dividing the entire search area into subregions and combining a (1 + 1)-EA with a greedy algorithm. Although relatively simple, EAGD provides a strong comparator, and our results showed that even the best EAGD performance did not surpass the average GA performance.

Overall, this study makes three main contributions:It frames SAR planning as a biomimetic optimization problem and applies a robust GA framework tailored to its structural complexities.It introduces a novel greedy initialization algorithm that enhances solution quality and convergence. This initialization approach was empirically validated to operate within strict SAR time limits, thus ensuring both effectiveness and practical deployability.It presents one of the few algorithmic studies that systematically integrate civilian SRU participation into SAR optimization, offering new directions for both computational research and real-world operational planning. Additionally, we incorporated statistical validation through *t*-tests, consistently obtaining *p*-values less than 0.001, which provides strong evidence of the significance of GA’s improvements over EAGD.

Future work will proceed along the following directions. (i) Expanded comparative baselines: We will conduct rigorous head-to-head evaluations against widely used single-objective metaheuristics (PSO, DE, SA, ACO, Tabu Search, VNS, GRASP, CMA-ES) and practical baselines including EAGD. All comparisons will follow a common evaluation protocol with identical fitness budgets, standardized parameter-tuning procedures, reproducible random seeds, and controlled computational resources. (ii) Multi-objective optimization for automated planning: We will formulate SAR planning as a multi-objective problem that jointly optimizes detection effectiveness, deployment time, fuel consumption, and operational risk. Using established algorithms (NSGA-II, MOEA/D), we will develop an automated pipeline that captures stakeholder preferences, generates Pareto-optimal solutions, and recommends plans based on decision-making criteria such as knee-point identification or reference-point methods. Comprehensive sensitivity analyses will quantify trade-offs across competing objectives. (iii) Adaptive parameter control with ablation: This work used fixed parameters. As a follow-up, we will adopt self-adaptive/rule-based control for crossover probability *p*, mutation rate, selection pressure, and the collision-avoidance threshold α, driven by population-diversity and convergence indicators, and report ablation that isolates each parameter/schedule’s effect. (iv) Dynamic/online planning: We will incorporate time-varying currents, winds, and visibility via predictive modeling and rolling-horizon reoptimization, together with uncertainty-aware schemes (scenario-based or distributionally robust optimization) to maintain performance under forecast error. (v) Learning-augmented acceleration: We will investigate surrogate-assisted operators that learn to approximate POD and collision/risk terms online (with active learning), and compare them with our current heatmap-based approach. (vi) Operational validation: Finally, we will conduct retrospective studies on historical SAR incidents and collaborate with maritime authorities to assess practicality, including mixed civilian–official deployments and operational constraints.

In addition, we highlight two further insights. First, collision-avoidance constraints were reformulated so that the altitude separation (initially set at 150 m following IAMSAR guidelines) is now a tunable parameter, broadening the applicability of our approach across different operational contexts. Second, our local optimization strategy was designed as a lightweight adjustment mechanism operating within one second, complementing the GA’s search without incurring significant runtime overhead. These refinements reinforce the practical feasibility of the proposed framework.

## Figures and Tables

**Figure 1 biomimetics-10-00588-f001:**
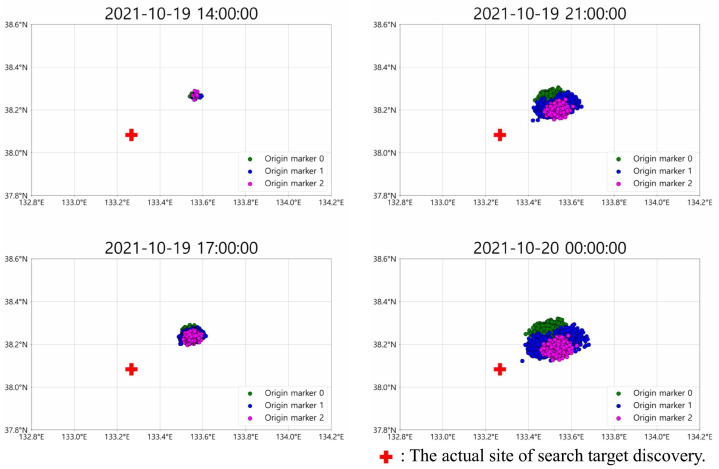
Example of particle drift prediction results (timestamps shown as YYYY-MM-DD hh:mm:ss).

**Figure 2 biomimetics-10-00588-f002:**
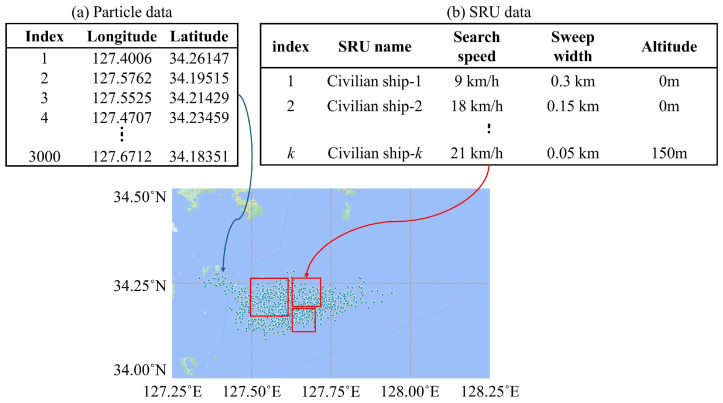
Example of input data: (**a**) particle data and (**b**) SRU data.

**Figure 3 biomimetics-10-00588-f003:**
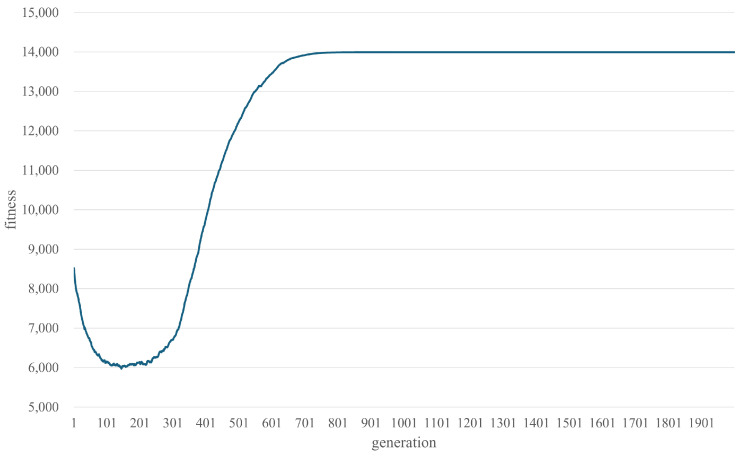
Convergence curve of the genetic algorithm on the Case 1, Scenario 3 dataset (coverage ratio 1.0).

**Figure 4 biomimetics-10-00588-f004:**
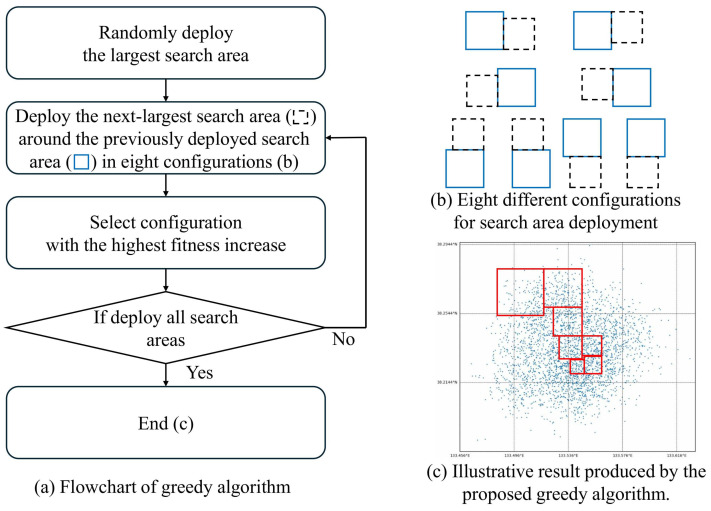
Greedy algorithm for search area deployment: (**a**) flowchart, (**b**) eight deployment configurations, and (**c**) illustrative deployment result (deployed search area (□)).

**Figure 5 biomimetics-10-00588-f005:**
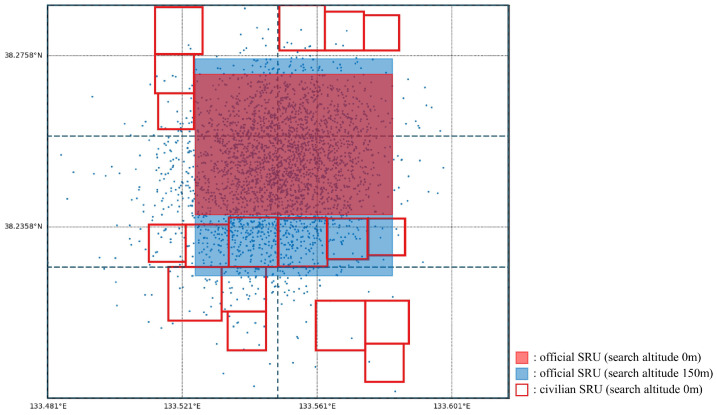
Deployment in Case 2, Scenario 1, coverage ratio 0.5: civilian SRUs (dark red) are assigned after official SRUs (light red) have been fixed.

**Figure 6 biomimetics-10-00588-f006:**
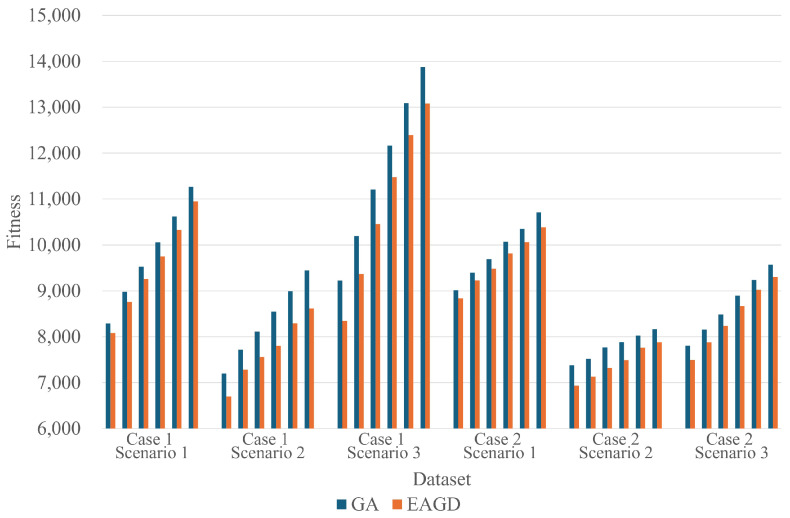
Comparison of average fitness across 30 repeated runs for each dataset.

**Table 1 biomimetics-10-00588-t001:** Comparison of GA and EAGD under different scenarios and ratios.

Case	Scenario	Ratio	GA			EAGD			*p*-Value
			Best	Ave ^1^	SD ^2^	Best	Ave ^1^	SD ^2^	
Case 1	Scenario 1	0.5	**8439.82**	8288.54	65.90	8101.64	8082.55	16.44	< 0.001
		0.6	**9129.64**	8976.14	76.38	8790.82	8759.99	11.19	<0.001
		0.7	**9691.76**	9524.65	65.28	9268.87	9261.94	4.09	< 0.001
		0.8	**10,200.80**	10,057.40	66.48	9765.89	9749.18	8.18	<0.001
		0.9	**10,752.20**	10,619.60	60.15	10,335.00	10,326.00	5.48	<0.001
		1.0	**11,460.50**	11,264.90	85.63	10,980.80	10,951.30	32.91	<0.001
	Scenario 2	0.5	**7244.98**	7198.26	23.75	6709.86	6697.30	4.48	<0.001
		0.6	**7799.73**	7719.80	23.88	7293.77	7282.74	19.30	<0.001
		0.7	**8226.80**	8112.64	48.06	7570.00	7557.90	18.14	<0.001
		0.8	**8633.88**	8545.65	35.58	7815.22	7803.50	12.01	<0.001
		0.9	**9056.57**	8990.16	34.08	8318.61	8290.85	9.08	<0.001
		1.0	**9539.08**	9443.64	37.74	8636.23	8617.21	5.65	<0.001
	Scenario 3	0.5	**9342.44**	9222.76	50.18	8375.92	8345.14	19.94	<0.001
		0.6	**10,308.60**	10,194.30	47.14	9378.72	9365.15	21.57	<0.001
		0.7	**11,303.40**	11,206.10	36.94	10,459.50	10,454.50	3.64	<0.001
		0.8	**12,302.60**	12,163.90	55.95	11,487.20	11,473.60	7.83	<0.001
		0.9	**13,205.20**	13,090.70	65.50	12,432.90	12,393.20	23.15	<0.001
		1.0	**13,992.40**	13,876.70	53.52	13,297.10	13,083.28	408.57	<0.001
Case 2	Scenario 1	0.5	**9073.02**	9014.23	22.10	8938.47	8837.08	50.75	<0.001
		0.6	**9447.08**	9396.65	24.32	9307.12	9228.98	53.77	<0.001
		0.7	**9746.78**	9691.09	23.59	9609.45	9482.29	79.57	<0.001
		0.8	**10,105.30**	10,068.50	23.47	9998.63	9813.68	89.56	<0.001
		0.9	**10,411.50**	10,347.60	24.93	10,246.10	10,060.40	70.48	<0.001
		1.0	**10,755.90**	10,708.20	23.92	10,650.10	10,383.20	177.57	<0.001
	Scenario 2	0.5	**7391.88**	7379.82	7.03	6936.58	6936.36	0.08	<0.001
		0.6	**7537.96**	7518.36	9.80	7130.54	7130.28	0.13	<0.001
		0.7	**7783.76**	7766.34	8.81	7318.68	7318.54	0.18	<0.001
		0.8	**7909.14**	7883.98	20.67	7492.18	7491.96	0.14	<0.001
		0.9	**8053.76**	8027.97	14.43	7760.88	7760.18	0.67	<0.001
		1.0	**8194.92**	8162.28	18.81	7883.20	7882.20	0.80	<0.001
	Scenario 3	0.5	**7843.45**	7805.61	22.83	7498.11	7495.10	4.41	<0.001
		0.6	**8192.54**	8153.44	18.02	7887.63	7877.57	5.31	<0.001
		0.7	**8533.79**	8486.09	22.59	8245.69	8234.85	5.70	<0.001
		0.8	**8941.63**	8896.42	20.15	8689.94	8670.18	8.79	<0.001
		0.9	**9285.73**	9237.69	26.80	9036.05	9022.18	9.47	<0.001
		1.0	**9638.11**	9569.24	31.44	9314.58	9302.30	7.04	<0.001

^1^ Ave: Average of fitness across 30 independent runs. ^2^ SD: Standard deviation of fitness across 30 independent runs. Values in **bold** denote the best result within each dataset (Case, Scenario, and Ratio).

## Data Availability

The original contributions presented in the study are included in the article, and further inquiries can be directed to the corresponding author.

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
