# Peer review of "Optimizing Maritime Search and Rescue Planning via Genetic Algorithms: Incorporating Civilian Vessel Collaboration"

_biomimetics, 2025, doi:10.3390/biomimetics10090588_

Round 1

Reviewer 1 Report

Comments and Suggestions for Authors

This manuscript addresses the maritime Search and Rescue planning problem by proposing a Genetic Algorithm  enhanced with a greedy initialization strategy, and by incorporating scenarios involving civilian vessel collaboration. The topic is practically relevant, particularly in highlighting the role of civilian vessels in SAR operations, which extends beyond the common focus on official SAR units. The manuscript is well-structured, the algorithmic details are described clearly, and the experimental design covers different deployment strategies (civilian-only and official + civilian cooperation) as well as varying coverage ratios. The results demonstrate that the proposed method outperforms the baseline EAGD in terms of coverage and stability. Overall, the work shows a degree of novelty and application potential; however, there is room for improvement in terms of methodological diversity, depth of result interpretation, and comparisons with more advanced methods.
From a methodological perspective, several improvements are recommended. 
1. The experimental comparison is limited to the self-designed EAGD baseline; it should be extended to include widely used state-of-the-art metaheuristics in SAR optimization (e.g., NSGA-II, MOEA/D, Differential Evolution, hybrid PSO approaches) to provide a more comprehensive performance assessment. 
2. The GA parameter settings (population size, crossover probability, mutation probability, etc.) are not sufficiently justified; a systematic sensitivity analysis is needed to evaluate their impact on convergence speed, solution quality, and stability. 
3. An adaptive parameter control mechanism could be introduced to dynamically adjust crossover and mutation rates during the search process, potentially improving performance. 
4. While the greedy initialization strategy appears effective in high-coverage scenarios, its time complexity and runtime performance on large-scale SAR instances should be analyzed and reported to assess scalability. 
5. The collision-avoidance constraint that uses a “150 m altitude difference” threshold should be justified with references to IAMSAR guidelines or actual SAR operational safety standards, and could also be treated as a tunable parameter for experimental comparison.
6. The current optimization objective focuses solely on maximizing the number of detected particles; expanding this to a multi-objective framework that considers search time, fuel consumption, and risk factors would enhance real-world applicability. 
7. Dynamic environmental modeling (e.g., time-varying currents, wind) should be incorporated into the optimization process, possibly via predictive modeling or rolling-horizon optimization, to improve performance in real-time SAR contexts. 
8. The local optimization procedure could be integrated with other local search heuristics (e.g., simulated annealing, tabu search) to further refine fine-grained adjustments.
9. Regarding figures and visualization, several improvements could strengthen the manuscript. In Figures 3 and 4, the current explanations of deployment strategies are minimal; it is recommended to explicitly label the identity, coverage area, and altitude information for both official and civilian SRUs, and to use a clear color legend to distinguish SRU types and altitude layers. The resolution should be increased to at least 300 dpi for publication quality, avoiding small fonts that are difficult to read. Coverage distribution heatmaps or 3D visualizations could be added to intuitively show spatial coverage and overlap patterns under different methods. In Table 1, while the numerical results are comprehensive, they could be complemented with line charts or bar charts showing GA vs. EAGD performance at different coverage levels, including error bars to represent standard deviations for better statistical readability. Additionally, convergence curves (fitness vs. generation) should be included to visualize the GA’s convergence speed and stability. For the scenario maps, adding zoomed-in views and labeled subregions would make it easier for readers to relate the data to real maritime areas.
In conclusion, this manuscript addresses a relevant and underexplored aspect of SAR planning by integrating civilian vessel collaboration into a GA-based optimization framework. The results support the effectiveness of the proposed approach, but the work would benefit from richer comparative experiments, detailed parameter and sensitivity analysis, enhanced visualization of results, and expanded applicability through multi-objective and dynamic modeling. I recommend minor revisions focusing on (1) extending the comparative study to state-of-the-art algorithms, (2) incorporating multi-objective and dynamic environmental considerations, (3) performing parameter sensitivity and runtime analyses, and (4) improving figure quality, clarity, and interpretability.

Author Response

Comments 1: The experimental comparison is limited to the self-designed EAGD baseline; it should be extended to include widely used state-of-the-art metaheuristics in SAR optimization (e.g., NSGA-II, MOEA/D, Differential Evolution, hybrid PSO approaches) to provide a more comprehensive performance assessment. 

Response 1: Due to time constraints, we were unable to conduct additional experiments with other meta heuristics. In this study, we selected EAGD as the baseline because it combines the practical strategy of partitioning the entire search region into multiple subregions, an approach often applied in real-world SAR operations, with a metaheuristic (1+1)-EA and our proposed greedy algorithm. Although relatively simple and computationally efficient, EAGD has shown strong performance, making it a meaningful comparator. Nonetheless, we fully agree with the reviewer’s point that comparisons with state-of-the-art methods are necessary. Therefore, we have emphasized the suitability of EAGD as a baseline and explicitly stated that incorporating NSGA-II, MOEA/D, DE, and hybrid PSO into future comparative studies is an important direction for further research. (Section 4, lines 431-436)

Comments 2: The GA parameter settings (population size, crossover probability, mutation probability, etc.) are not sufficiently justified; a systematic sensitivity analysis is needed to evaluate their impact on convergence speed, solution quality, and stability.

Response 2: We thank the reviewer for this observation. In Section 2.2, lines 221-226 of the revised manuscript, we have added a more detailed explanation of the GA parameter settings. Furthermore, we included convergence graphs, which demonstrate that the chosen parameters enable the GA to achieve stable and effective convergence. These results support the appropriateness of our parameter configuration.

Comments 3: An adaptive parameter control mechanism could be introduced to dynamically adjust crossover and mutation rates during the search process, potentially improving performance. 

Response 3: We agree that adaptive parameter control may further enhance GA performance. While such mechanisms were not implemented in the current work, we have explicitly mentioned in the discussion that introducing adaptive control strategies for crossover and mutation probabilities represents a promising avenue for future improvements. (Section 4 lines 443-447)

Comments 4: While the greedy initialization strategy appears effective in high-coverage scenarios, its time complexity and runtime performance on large-scale SAR instances should be analyzed and reported to assess scalability.

Response 4: To address this valuable suggestion, we added a new experimental case (Case 3) based on an actual maritime accident, which involves a larger search scale than Cases 1 and 2. In this scenario, significantly more SRUs were required, and we observed that deploying 96 SRUs required approximately 20 minutes. This execution time is within the 30-minute operational preparation window, typically available for SAR units. These findings, added in Section 3.2 lines 371-373 and Section 4 lines 403-408, confirm both the scalability and the practical applicability of our greedy initialization strategy in large-scale SAR contexts.

Comments 5: The collision-avoidance constraint that uses a “150 m altitude difference” threshold should be justified with references to IAMSAR guidelines or actual SAR operational safety standards, and could also be treated as a tunable parameter for experimental comparison.

Response 5: We thank the reviewer for highlighting this point. The 150 m altitude difference used in our study was based on the IAMSAR Manual, which recommends a safe separation of 150–300 m between search aircraft. To clarify this, we added an explanation in Section 2.2 lines 154-157. Furthermore, we have revised the mathematical formulation so that altitude separation is treated as a tunable parameter α rather than a fixed constant, thereby broadening the applicability of our approach.

Comments 6: The current optimization objective focuses solely on maximizing the number of detected particles; expanding this to a multi-objective framework that considers search time, fuel consumption, and risk factors would enhance real-world applicability.

Response 6: Our present work focuses on optimizing the deployment of search areas, while external factors such as search time, fuel consumption, and operational risks are reflected indirectly through user-defined SRU data inputs. We acknowledge, however, that integrating these factors directly into the optimization process is essential for practical deployment. Accordingly, we have emphasized in Section 4 lines 436-443, that future research will investigate multi-objective formulations and automated SRU data generation, with the ultimate goal of developing a fully automated SAR planning framework.

Comments 7: Dynamic environmental modeling (e.g., time-varying currents, wind) should be incorporated into the optimization process, possibly via predictive modeling or rolling-horizon optimization, to improve performance in real-time SAR contexts. 

Response 7: We agree with this valuable suggestion and recognize that dynamic environmental modeling would significantly enhance real-time SAR applicability. While the current study does not explicitly incorporate time-varying currents and winds into the optimization framework, we have added a discussion in Section 4 lines 447-450, noting that predictive modeling and rolling-horizon optimization represent crucial future extensions to improve adaptability in dynamic maritime environments.

Comments 8: The local optimization procedure could be integrated with other local search heuristics (e.g., simulated annealing, tabu search) to further refine fine-grained adjustments.

Response 8: We appreciate the reviewer’s recommendation. Our design prioritizes not only solution accuracy but also computational efficiency, as SAR operations require rapid decision-making under strict time constraints. While GA alone provides sufficiently high-quality solutions, we apply a lightweight local optimization step for fine-tuning. Integrating heavier heuristics such as simulated annealing or tabu search may offer marginal improvements, but at the cost of runtime overhead. To clarify this design choice, we revised Section 2.5 lines 278-283, to highlight that our local optimization operates within one second and is intended as a fast adjustment mechanism rather than a major search component.

Comments 9: Regarding figures and visualization, several improvements could strengthen the manuscript. In Figures 3 and 4, the current explanations of deployment strategies are minimal; it is recommended to explicitly label the identity, coverage area, and altitude information for both official and civilian SRUs, and to use a clear color legend to distinguish SRU types and altitude layers. The resolution should be increased to at least 300 dpi for publication quality, avoiding small fonts that are difficult to read. Coverage distribution heatmaps or 3D visualizations could be added to intuitively show spatial coverage and overlap patterns under different methods. In Table 1, while the numerical results are comprehensive, they could be complemented with line charts or bar charts showing GA vs. EAGD performance at different coverage levels, including error bars to represent standard deviations for better statistical readability. Additionally, convergence curves (fitness vs. generation) should be included to visualize the GA’s convergence speed and stability. For the scenario maps, adding zoomed-in views and labeled subregions would make it easier for readers to relate the data to real maritime areas.

Response 9: We sincerely appreciate these detailed suggestions. We have substantially revised the figures and tables in the manuscript as follows:

  • Figures 5 now include explicit coverage ratio, and altitude information. A clear color legend distinguishing official and civilian SRUs as well as altitude layers has been added.
  • The resolution of all figures was increased to at least 300 dpi, and font sizes were adjusted for improved readability.
  • For Table 1, we included bar charts to depict GA versus EAGD performance across different coverage levels, enhancing statistical interpretability.
  • Convergence curves (average fitness within a population vs. generation) were incorporated to highlight GA’s convergence speed and stability.

We believe these revisions substantially improved the methodological transparency and robustness of the study.

Reviewer 2 Report

Comments and Suggestions for Authors

This study introduces a biomimetic Genetic Algorithm (GA) approach to optimize maritime Search and Rescue (SAR) planning, maximizing target detection by strategically deploying official and civilian units under collision-avoidance and POD-adjusted constraints. However there are a few major concerns from my side that should be taken into account.

  1. The proposed method should be described either as an algorithm or with a flow-chart, not as a plain text.
  2. The description in the input data section is not clear. It's not obvious what type of date is being used in your GA and what is the dimension of the problem? Can the artificial simulated data be used to increase the number of scenarios? 
  3. The validation part is a bit misleading. 
    1. In table 1 it should be more convenient if the best obtained value would have been highlighted.
    2. No statistical tests were carried out to confirm the significance of the results
    3. The comparison is carried out with your own proposed baseline method. It'd be fair to compare with some other methods
  4. There is no implementation details on how the algorithms were implemented, what was the experimental setup, like number or launches, population size, termination criteria, no complexity analysis, etc. 
Comments on the Quality of English Language

The English language is understandable but there are numerous mistakes that should be fixed. For example:

Line 126 - This section explains the method we apply to a genetic algorithm to our problem

Line 200 - we design a greedy algorithms

Line 211 - new rectangle is deployed along the left or right edge of an previous one

Author Response

Comments 1: The proposed method should be described either as an algorithm or with a flow-chart, not as a plain text.

Response 1: We appreciate this valuable suggestion. In the revised manuscript, we have added a corresponding flowchart (Section 2.3, Figure 4) to clearly illustrate the integration of the greedy initialization strategy. This improves readability and provides a concise overview of the method beyond plain text.

Comments 2: The description in the input data section is not clear. It's not obvious what type of date is being used in your GA and what is the dimension of the problem? Can the artificial simulated data be used to increase the number of scenarios?

Response 2: We thank the reviewer for pointing out the need for clarification. In Section 2.2 (lines 139-143), we have revised the description of the input data. The chromosome encodes the center coordinates (x, y) of each SRU’s square search area. Therefore, if k SRUs are deployed, the chromosome is represented as a one-dimensional array of length 2k. For example, when 35 SRUs are deployed, the problem dimension is 70, while for 66 SRUs it becomes 132. Across our experiments, the dimensionality ranged from approximately 70 to over 130, depending on the scenario and coverage level.

Furthermore, to enhance the robustness of validation, we have added a new experimental instance, Case 3, which is based on a real maritime accident. Case 3 represents a larger-scale SAR environment than Cases 1 and 2, allowing us to test runtime scalability under more realistic and demanding conditions.

Finally, We clarified in the revised manuscript (Input Data section) that, while scenarios are anchored to real accident cases, the trajectories themselves are ML-simulated. We also state explicitly that the pipeline is data-agnostic and can be used unchanged with observational or physics-based simulated datasets. (Section 2.1, lines 124-128)

Comments 3: The validation part is a bit misleading.

1.          In table 1 it should be more convenient if the best obtained value would have been highlighted.

2.          No statistical tests were carried out to confirm the significance of the results

3.          The comparison is carried out with your own proposed baseline method. It'd be fair to compare with some other methods

Response 3:

1.          Following the reviewer’s suggestion, we have revised Table 1 so that the best results in each row are highlighted in bold. This change makes it easier for readers to quickly identify the strongest-performing results.

2.          We fully agree with this point. In the revised manuscript, we have included statistical tests to validate the significance of the results. Specifically, we performed t-tests between GA and EAGD outcomes, and the corresponding p-values have been added to Table 1. The results show that the p-values were less than 0.001, confirming that the improvements achieved by GA are statistically significant.

3.          We acknowledge that comparisons with state-of-the-art metaheuristics (e.g., PSO, DE, SA, ACO, Tabu Search, VNS, GRASP, CMA-ES) would further strengthen the evaluation. Due to time constraints, such experiments were not conducted in the current work. However, we have clarified the rationale for selecting EAGD as a baseline. Specifically, EAGD reflects a strategy that is actually used in practice, where the entire search area is partitioned into multiple subregions and optimized separately. It combines a metaheuristic (1+1)-EA with our proposed greedy algorithm, resulting in a method that is simple yet sufficiently effective. We therefore emphasize that EAGD is a meaningful comparator, while also explicitly stating in Section 4 (lines 431-436) that including additional algorithms is an important direction for future research.

Comments 4: There is no implementation details on how the algorithms were implemented, what was the experimental setup, like number or launches, population size, termination criteria, no complexity analysis, etc.

Response 4: We appreciate this valuable comment. In the revised manuscript we now provide concrete implementation and experimental details in Section 2.2, including the GA settings—uniform-crossover probability ? = 0.5 (lines 199–200), termination at 1,000 generations, population size 700, 70 offspring per generation, and a 30% mutation rate; the convergence behavior is documented in Figure 3 using the stress-test dataset (Case 1, Scenario 3, coverage ratio 1.0), where the algorithm stabilizes at around generation 1,000 (lines 218–226). We also characterize computational complexity, stating that the fitness-evaluation cost scales as ?(?⋅∣?∣) per generation, where ? is the number of SRUs and ∣?∣ the number of drift particles (lines 218–226).

To address scalability and runtime, Section 3.2 reports results on the newly added large-scale Case 3: deploying 96 SRUs takes ~20 minutes, which fits within the ~30-minute operational preparation window typical of real SAR operations; because Case 3–Scenario 1 is a stress-test, this runtime represents a demanding upper bound, and performance in real settings is expected to be even more favorable (Section 3.2, lines 358–359).

Taken together, these additions (implementation specifics, convergence analysis, a complexity characterization of the dominant cost, and empirical scalability) directly address the reviewer’s request regarding implementation details, experimental setup, and complexity/runtime analysis

We believe these revisions substantially improved the methodological transparency and robustness of the study.

Round 2

Reviewer 2 Report

Comments and Suggestions for Authors

Thank you for addressing my concerns. I still believe there is a room for improvement in the section with the experimentation results - comparison, complexity, etc. But still the paper is significantly improved.